# Safety and Efficacy of Outpatient Treatments for COVID-19: Real-Life Data from a Regionwide Cohort of High-Risk Patients in Tuscany, Italy (the FEDERATE Cohort)

**DOI:** 10.3390/v15020438

**Published:** 2023-02-05

**Authors:** Tommaso Manciulli, Michele Spinicci, Barbara Rossetti, Roberta Maria Antonello, Filippo Lagi, Anna Barbiero, Flavia Chechi, Giuseppe Formica, Emanuela Francalanci, Mirco Alesi, Samuele Gaggioli, Giulia Modi, Sara Modica, Riccardo Paggi, Cecilia Costa, Alessandra Morea, Lorenzo Paglicci, Ilaria Rancan, Francesco Amadori, Agnese Tamborrino, Marta Tilli, Giulia Bandini, Alberto Moggi Pignone, Beatrice Valoriani, Francesca Montagnani, Mario Tumbarello, Pierluigi Blanc, Massimo Di Pietro, Luisa Galli, Donatella Aquilini, Antonella Vincenti, Spartaco Sani, Cesira Nencioni, Sauro Luchi, Danilo Tacconi, Lorenzo Zammarchi, Alessandro Bartoloni

**Affiliations:** 1Dipartimento di Medicina Sperimentale e Clinica, Università degli Studi di Firenze, 50121 Firenze, Italy; 2SOD Malattie Infettive e Tropicali, Azienda Ospedaliero-Universitaria Careggi, 50134 Firenze, Italy; 3UOC Malattie Infettive, Ospedale Misericordia, 58100 Grosseto, Italy; 4SOC Malattie Infettive, Ospedale San Jacopo, 51100 Pistoia, Italy; 5UO Malattie Infettive, Ospedale Santo Stefano, 59100 Prato, Italy; 6UO Malattie Infettive, Ospedale Santa Maria Annunziata, 50012 Firenze, Italy; 7SOC Malattie Infettive ed Epatologia, Ospedale San Luca, 55100 Lucca, Italy; 8UO Malattie Infettive, Ospedali Riuniti di Livorno, 57124 Livorno, Italy; 9UO Malattie Infettive, Ospedale San Donato, 52100 Arezzo, Italy; 10Dipartimento di Biotecnologie Mediche, Università degli Studi di Siena, 53100 Siena, Italy; 11UO Malattie Infettive, Ospedale Apuane, 54100 Massa, Italy; 12Dipartimento Scienze della Salute, Università degli Studi di Firenze, 50139 Firenze, Italy; 13UOC Malattie Infettive e Tropicali, Azienda Ospedaliero Universitaria Senese, 53100 Siena, Italy; 14UO Malattie Infettive, Azienda Ospedaliero-Universitaria “Meyer”, 50139 Firenze, Italy

**Keywords:** SARS-CoV-2, COVID-19, sotrovimab, nirmatrelvir/ritonavir, molnupiravir, remdesivir

## Abstract

Early COVID-19 treatments can prevent progression to severe disease. However, real-life data are still limited, and studies are warranted to monitor the efficacy and tolerability of these drugs. We retrospectively enrolled outpatients receiving early treatment for COVID-19 in 11 infectious diseases units in the Tuscany region of Italy between 1 January and 31 March 2022, when Omicron sublineages BA.1 and BA.2 were circulating. Eligible COVID-19 patients were treated with sotrovimab (SOT), remdesivir (RMD), nirmatrelvir/ritonavir (NRM/r), or molnupiravir (MOL). We gathered demographic and clinical features, 28-day outcomes (hospitalization or death), and drugs tolerability. A total of 781 patients (median age 69.9, 66% boosted for SARS-CoV-2) met the inclusion criteria, of whom 314 were treated with SOT (40.2%), 205 with MOL (26.3%), 142 with RMD (18.2%), and 120 with NRM/r (15.4%). Overall, 28-day hospitalization and death occurred in 18/781 (2.3%) and 3/781 (0.3%), respectively. Multivariable Cox regression showed that patients receiving SOT had a reduced risk of meeting the composite outcome (28-day hospitalization and/or death) in comparison to the RMD cohort, while no significant differences were evidenced for the MOL and NRM/r groups in comparison to the RMD group. Other predictors of negative outcomes included cancer, chronic kidney disease, and a time between symptoms onset and treatment administration > 3 days. All treatments showed good safety and tolerability, with only eight patients (1%) whose treatment was interrupted due to intolerance. In the first Italian multicenter study presenting real-life data on COVID-19 early treatments, all regimens demonstrated good safety and efficacy. SOT showed a reduced risk of progression versus RMD. No significant differences of outcome were observed in preventing 28-day hospitalization and death among patients treated with RMD, MOL, and NRM/r.

## 1. Introduction

Therapeutic options for the early phase of coronavirus disease 2019 (COVID-19) have been sought since the start of the pandemic. In the last year, several compounds have been licensed for the treatment of patients with recent symptoms onset [1]. These include monoclonal antibodies (mAbs); the oral antivirals nirmatrelvir/ritonavir (NRM/r; an inhibitor of the main protease, also called 3-chymotrypsin-like protease, of severe acute respiratory syndrome coronavirus 2 (SARS-CoV-2)) and molnupiravir (MOL; an inhibitor of the RNA-dependent RNA polymerase of SARS-CoV-2); and the intravenous antiviral remdesivir (RMD, another inhibitor of the RNA-dependent RNA polymerase of SARS-CoV-2) [2]. Based on the results of clinical trials, most international guidelines issued recommendations which prioritize the use of NRM/r (relative risk [RR] reduction 88% for hospitalization or death) or RMD (RR reduction 87%) over MOL (RR reduction 30%) for the treatment of COVID-19 patients who do not require hospitalization or supplemental oxygen [1,3]. The mAbs group has included several molecules, often used in combination to achieve viral neutralization. The clinical trials testing mAbs use in outpatients have been carried out in patients carrying the Alpha variant [4,5,6], and over time some of these drugs became inefficacious as SARS-CoV-2 mutated, generating new variants [7]. Sotrovimab (SOT), a recombinant human monoclonal antibody against SARS-CoV-2, which obtained US Food and Drug Administration Emergency Use Authorization for the treatment of high-risk outpatients with mild-to-moderate COVID-19 in May 2021, has been shown to have lower neutralizing activity against Omicron BA.1 than against the ancestral strain and other variants of concern, even less neutralizing activity against Omicron BA.2, and lost inhibitory capability against BA.4 and BA.5 [8,9]. Monoclonal antibodies that have maintained activity against BA.4/5 include bebtelovimab [1,7,10], currently not approved in Europe, and the combination of tigaxevimab and cilgavimab [7,11], approved for both pre-/post-exposure prophylaxis and early treatment of immunocompromised patients at increased risk of severe COVID. However, the clinical trials for most of these drugs were carried out in the pre-Omicron era, and among non-vaccinated subjects [4,12]. The introduction of anti-SARS-CoV-2 vaccines caused a drastic reduction in COVID-19 severity and mortality [13,14,15], and it also changed the susceptibility of the target population to the virus. As such, continued surveillance of the efficacy and effectiveness of the compounds used to treat patients at high risk of severe disease is needed. To date, few studies on the efficacy and tolerability of these drugs under real-world conditions, characterized by new variants of concern (VOCs) and large vaccination coverage, have been published. For instance, a recent observational study from Israel showed that NRM/r was able to reduce hospitalization rates and deaths in treated versus untreated subjects [16,17]. Early treatment with either MOL or NRM/r was confirmed to reduce the risks of mortality and in-hospital disease progression in comparison with untreated controls in a large cohort of patients (mostly unvaccinated) in Hong Kong, during the wave of SARS-CoV-2 Omicron subvariant BA.2.2, as NRM/r was additionally associated with a reduced risk of hospitalization. Other observational studies have confirmed the promising results of NRM/r [18,19,20], while data on the performance of MOL appears to be less clear-cut, as one study has shown that untreated patients had similar outcomes to those receiving MOL [19,21]. However, available studies suffer from limitations arising from their retrospective, observational nature. Since the start of January 2022, different regimens for early treatment of COVID-19 have been available in Italy (SOT, RMD, and MOL), while NRM/r was made available at the start of February 2022. As such, these drugs have been employed in the “Omicron era” of COVID-19 and used on subjects deemed at risk of severe disease by the Italian National Drug Agency (AIFA), with at-risk conditions including chronic diseases such as hypertension with organ damage, chronic kidney disease, chronic heart disease, cancers, chronic lung disease, and immunosuppression, as well as an age over 65 years [22].

We present data on the safety and efficacy of the four outpatient regimens available in Italy (SOT, RMD, NRM/r, and MOL), obtained from a multicenter study conducted in 11 infectious diseases units operating in the Tuscany region of Italy.

## 2. Materials and Methods

### 2.1. Patient Population

We retrospectively retrieved data on patients treated at the outpatient services of 11 infectious diseases units in Tuscany, Italy, between 1 January 2022 and 31 March 2022. Patients were considered eligible if: (i) they had received SOT, RMD, NRM/r, or MOL; (ii) they were treated in an outpatient setting; (iii) they had at least one risk factor according to AIFA criteria; and (iv) they were classified as having mild or moderate COVID-19 infection according to WHO criteria. Symptoms that allowed for treatment were defined by AIFA criteria and included fever, malaise, smell or taste disturbances, chills, dyspnea, sore throat, headache, myalgia, and GI symptoms. On the other hand, patients were excluded if they were: (i) hospitalized for reasons other than COVID-19 at the time of treatment, (ii) without a risk factor for severe COVID-19 according to AIFA criteria, and/or (iii) asymptomatic or suffering from a severe or critical disease. Children from a pediatric infectious diseases center were included, in light of a recent position paper by the Italian Society of Pediatrics recommending early treatment options for children at risk of COVID-19 progression [23].

### 2.2. Data Collection

Collected data included demographic information (sex at birth, age), data on risk factors for COVID-19 progression according to the AIFA criteria: age > 65 years, hypertension with organ damage, chronic heart disease, chronic kidney disease (CKD), chronic lung disease, chronic liver disease, immunosuppression (either congenital or iatrogenic), oncological patients including those with blood and solid cancers undergoing active treatment, and obesity, defined by a body mass index [the weight in kilograms divided by the square of the height in meters] ≥30. Among these conditions, no weights were attributed by AIFA to regulate the prescription of anti-SARS-CoV-2 early treatments. We also collected information on vaccination status (defined as one-dose, full-cycle, or boosted regardless of the type of vaccine used, as information on vaccine type was not readily available), date of symptoms onset, date of treatment administration, latency between symptoms onset and treatment administration (defined as the number of days between the first day of symptoms and the day of treatment start). Information regarding the referral channel to outpatient services was also collected (hospital-based specialist, family doctor, doctor part of the special units set up for COVID-19 at-home management, direct referral from emergency department doctor).

Outcome measures included treatment completion, side effects (patient-reported intolerance to drug altering the course of treatment, allergic reaction), and hospitalization or death due to COVID-19 progression. A composite outcome consisting of death and/or hospitalization was also created.

Information on the outcome at 28 days was captured through a standardized questionnaire. Information on outcome measures was entered into the database at this time. Data were collected using REDCap 8.11.6. (Project REDCap, USA).

### 2.3. Data Analysis

We analyzed data for patients who had received treatment, using available information on the occurrence of hospitalization and death. Data were analyzed with STATA 17.0 (STATACorp, College Station, TX, USA). Continuous variables were reported as medians and interquartile ranges; categorical variables were reported as absolute counts and proportions. A chi-square test was used to test for differences in categorical variables. The Kruskal–Wallis test was used to test for differences in continuous variables among the treatment groups. A survival analysis between different treatment groups was carried out using Kaplan–Meier curves and the log-rank test. The average time at risk for an event was computed as the time to the first event or day 28, whichever was earlier. Multivariable Cox regression was performed to identify independent predictors of composite outcome (28-day hospitalization and/or death related to COVID-19), calculated as hazard ratios (HR, 95%CI). Moreover, given the non-randomized assignment to the four treatment groups, a propensity score (PS) analysis using inverse probability of treatment weighting (IPTW) was performed to assess the average treatment effect (ATE) of SOT, MOL, and NRM/r in comparison to RMD. Inverse probability of treatment weighting uses weights based on the propensity score to create a synthetic sample in which the distribution of measured baseline covariates is independent of treatment assignment [24]. The following covariates were included to generate the PS: sex; age; chronic comorbidities, such as obesity, chronic kidney disease, chronic heart disease, chronic obstructive pulmonary disease, cancer, cognitive impairment, diabetes, and immunosuppression; smoking habit; vaccination status, categorized as ‘not vaccinated’ (none or incomplete primary schedule) or ‘vaccinated’ (complete primary schedule +/− booster dose); and latency between symptoms onset to antiviral administration, categorized as ≤3 or >3 days. We arbitrarily decided to enter RMD as a reference variable, since it was the group with the highest number of events (hospitalization and/or death). Standardized differences were used to compare balance in baseline covariates between the four groups before and after weighing by the inverse probability of treatment.

### 2.4. Ethics

The study was performed in accordance with the ethical principles of the Declaration of Helsinki and with the International Conference for Harmonization Good Clinical Practice guidelines.

## 3. Results

In the study period, 921 patients received early treatment for mild-to-moderate COVID-19 within the 11 ID units involved. Of these, 140 (15.2%) did not meet the inclusion criteria and were excluded (Appendix A). Of the 781 included patients (50% female, median age 66.9 years, IQR 52.3–77.9), 314 (40.2%) received SOT, 142 (18.2%) received RMD, 205 (26.3%) received MOL, and 120 (15.4%) received NRM/r. In brief, patients receiving SOT (47% female) had the lowest median age (64.7 years, IQR 50.2–77.7) and included the highest percentage of non-vaccinated (20%, 64/314) and immunocompromised people (51%, 159/314). Patients in the MOL group (42% female, median age 68.9, IQR 57.3–79.9) were the oldest, and had the highest frequency of obese people (30%, 61/205). The RMD group (59% female, median age 67.4 years, IQR 52–78.9) had the highest percentage of smokers (31%, 44/142). The NRM/r group (57.5% female, median age 66.8 years, IQR 50.3–75.6) included the highest percentage of people fully vaccinated ± a booster dose (97%, 116/120). The baseline characteristics of the study population, divided into the four treatment groups, are fully reported in Table 1.

Most patients were referred to the prescribing centers either by their general practitioners (*n* = 327, 42.5%) or by territorial medical units for the care of COVID-19 (*n* = 258, 33.5%); the remainder of patients were referred either by other specialists (*n* = 111, 14.4%) or by the emergency department (*n* = 45, 5.8%). In 29 cases (3.8%), the patients had direct contact with the ID specialist. No information was available for 11 patients. We found that latency between symptoms onset and treatment was significantly higher for patients treated with parenteral drugs, i.e., SOT (median 4 days, IQR 3–5) and RMD (median 4 days, IQR 2–5) compared to oral antivirals MOL (median 3 days, IQR 2–4) and NRM/r (median 3 days, IQR 2–3) (*p* = 0.001).

### Outcome Data

Deaths occurred in one patient in the SOT group (0.3%) and in two patients in the RMD group (1.4%). No deaths occurred in the MOL and NRM/r groups. Eighteen patients were hospitalized due to COVID-19 progression: five (1.6%) in the SOT group, seven (4.9%) in the RMD group, four (1.9%) in the MOL group, and three (2.5%) in the NRM/r group.

Patients receiving treatment > 3 days from symptoms onset had a higher risk of meeting the composite endpoint of death or hospitalization (12/317, 3.8%) in comparison with those who started the treatment ≤ 3 days from symptoms onset (4/464, 1.3%, *p* = 0.023).

Kaplan–Meier survival curves for the composite outcome of each treatment group are reported in Figure 1.

The average time at risk for an event was 27.35 days for RMD (standard error [SE] 0.27), 27.74 days for SOT (SE 0.13), 27.67 days for MOL (SE 0.19), and 27.44 days for NRM/r (SE 0.32). Head-to-head comparison of survival curves between each treatment group showed significant differences only for RMD vs. SOT (difference in the cumulative percentage of patients with COVID-19–related hospitalization or death through day 28 was 3.3%, 95%CI −0.5–7.2%; *p*-value = 0.039). No statistical differences between other study groups were observed in the survival analysis.

Multivariable analysis performed by Cox regression showed that patients receiving SOT had a lower risk of meeting the composite outcome compared to patients in the RMD group (HR 0.14, 95%CI 0.03–0.56, *p* = 0.005), while no significant differences were evidenced between the RMD group and the MOL (HR 0.43, 95%CI 0.09–1.96, 0.273) and NRM/r groups (HR 0.51, 95%CI 0.11–2.28, 0.374). Predictors of hospitalization and/or death included a latency >3 days between symptoms onset and treatment administration (HR 1.41, 95%CI 1.07–1.85, *p* = 0.013), chronic kidney disease (HR 5.01, 95%CI 1.30–19.3, *p* = 0.019), and cancer (HR 3.11, 95%CI 1.07–9.09, *p* = 0.038), while a history of chronic heart disease resulted in a protective factor (HR 0.24, 95%CI 0.07–0.80, *p* = 0.020). Complete results of the Cox regression analysis are shown in Figure 2.

On IPTW-adjusted PS analysis, a trend in favor of SOT versus RMD was observed (ATE −0.04, 95%CI −0.07–0.002, *p* = 0.063), while no significant differences emerged when comparing the RMD group with the MOL (−0.01, 95%CI −0.07–0.04, *p* = 0.659) and NRM/r (0.00, 95%CI −0.07–0.07, *p* = 0.983) groups. Analysis of standardized differences showed good balance in baseline covariates between the four groups before and after weighting by IPTW (Appendix A).

Drug intolerance was reported by 29 patients (4%), including eight cases leading to drug discontinuation. Intolerance was reported by 5% in the MOL (10/205) and NRM/r (6/120) groups, 4% in the RMD group (5/142), and 3% in the SOT group (8/314). Discontinuation occurred only in the MOL (*n* = 5, 2.5%) and RMD groups (*n* = 3, 2.1%).

## 4. Discussion

Outpatient treatments for COVID-19 patients play a crucial role in the prevention of disease progression to severe forms in patients at high risk of poor outcomes [1]. However, continued evaluation of the safety and efficacy of these treatments is warranted, as new SARS-CoV-2 VOCs will emerge, and population susceptibility will change due to previous exposure and vaccine administration [13,14].

This study represents, to our knowledge, the largest multicenter report of real-life data from Italy, where the prescription of antivirals and monoclonal antibodies has been subject to strict regulation since their introduction in March 2021 [25]. Regulatory trials for all available compounds either were carried out before SARS-CoV-2 vaccine rollout or excluded vaccinated subjects [4,5,26]. Moreover, most trials did not focus on high-risk subjects, except for one trial on the use of SOT [27]. Our study population is largely representative of patients that are currently at greater risk for COVID-19, i.e., elderly patients older than 65 years old, with multiple comorbidities predisposing to severe COVID-19, albeit mostly vaccinated against COVID-19 [1,21,28,29].

All drugs showed low rates of hospitalization and/or death due to COVID-19 progression, in line with results from previous studies [16,19,30]. Multivariable analysis suggested a possible advantage in the use of SOT in comparison with RMD, while no significant differences were observed among the three antiviral agents (RMD, MOL, and NRM/r).

This study was conducted in the so-called “Omicron era”: in Tuscany, the Omicron lineage B.1.1.529 was responsible for around 90% of new infections at the beginning of 2022, and reached 100% at the end of the study period in March 2022, when BA.1 and BA.2 were at 52% and 47%, respectively [31,32]. The Omicron variant has been associated with a reduced risk of hospitalization and death in the general population compared to the Delta variant, although significant variation has been observed by age [33]. Moreover, immunocompromised patients infected with the Omicron variant remain at high risk of severe outcomes, as observed in a prospective cohort of 114 solid organ transplant recipients, patients on anti-CD20 therapy, and allogenic hematopoietic stem-cell transplantation recipients, one of whom died and 23 (20%) of whom required hospital admission for a median of 11 days [34].

Chronic kidney disease and cancer were confirmed to be predictors of severe outcomes in patients with COVID-19, regardless of the use of early treatment against SARS-CoV-2. Conversely, chronic heart disease was a predictor of positive outcomes. Both severe CKD and cancer emerged as higher-risk comorbidities compared with other conditions, such as old age, chronic heart diseases, metabolic disorders, or isolated hypertension, in a multicenter cohort study carried out in Shanghai, China, during the 2022 Omicron wave [35]. Moreover, CKD patients have reduced treatment options, since the use of NRM/r and RMD is contraindicated in patients with severe renal function impairment, i.e., a glomerular filtration rate less than 30 mL/min, limiting the choice to MOL and/or mAbs.

It should be highlighted that SOT is not effective against Omicron BA.4 and BA.5, the currently dominant subvariants [7,10]. However, we decided to include SOT patients in the analysis, considering that real-life data on this compound could still be informative for the future use of other antibodies. On the other hand, no report of SARS-CoV-2 resistance to RMD and/or oral antivirals (MOL and NRM/r) has emerged to date, and susceptibilities of BA.4 and BA.5 VoCs to the three compounds were similar to those of the ancestral SARS-CoV-2 strain [36].

All patients were prescribed the drugs within a relatively short period of time from symptoms onset, within a median 3 days for oral antivirals and 4 days for RMD and SOT, according to AIFA criteria [22]. It is worth noting that a time from symptoms onset to treatment administration longer than 3 days was a predictor of a negative outcome in our population. This finding, along with the absence of significant difference in the outcomes, supports the use of oral compounds in situations where logistics issues may delay administration of parenteral drugs [25].

Moreover, the four treatments appear to be acceptably safe in terms of adverse events, which ranged from 3 to 5% of patients, similar to those found in regulatory clinical trials and other real-life studies [16,19,30].

The main limitation of this study is its retrospective design and the non-randomized assignment to the four treatment groups. Different distributions of patients’ features and comorbidities across groups are related to the different drug characteristics and/or may reflect specific attitudes of prescribers. For example, the higher frequency of chronic kidney disease in the SOT group can be explained by drug pharmacokinetics (i.e., no potential for nephrotoxicity, unlike both RMD and NRM/r). The excess of unvaccinated and immunocompromised people in the SOT group is likely to reflect a greater confidence in this compound for the frailest patients. Furthermore, NRM/r was not available until mid-February in Italy.

Furthermore, we did not collect and analyze data about COVID-19 symptoms, and we did not investigate the potential correlation between clinical manifestation and COVID-19 severity. Likewise, we could not retrieve data on the full immunization schedules of all participants: the immunization campaign in Italy has used different combinations of vaccines since its start in 2020 [37], and we cannot exclude the possibility that such variables might influence COVID-19 outcomes and drug tolerability. However, an exhaustive analysis of the role of these variables in COVID-19 patients was beyond the scope of our study. Another limitation is that we did not collect data on the time to viral clearance, nor on the presence of rebound infection, which has recently been reported after the administration of NRM/r and MOL [18,38].

## 5. Conclusions

In conclusion, this study represents one of the first efforts at real-life data collection on COVID-19 outpatient treatment options. We observed a low incidence of adverse events and negative outcomes with all currently used treatments, and we confirmed the paramount importance of the administration timing of early therapies against COVID-19.

## Figures and Tables

**Figure 1 viruses-15-00438-f001:**
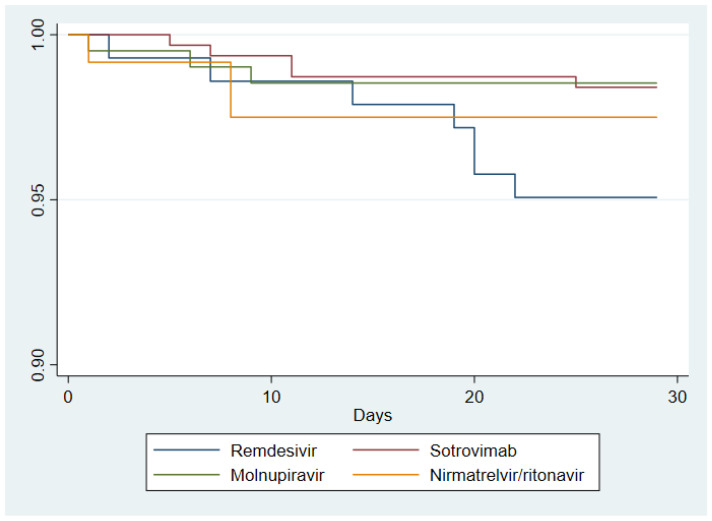
Kaplan–Meier survival curves of patients assigned to the four treatment groups (SOT *n* = 314; MOL *n* = 205; RMD *n* = 142; NRM/r *n* = 120). The failure event is a composite outcome of 28-day hospitalization and/or death related to COVD-19.

**Figure 2 viruses-15-00438-f002:**
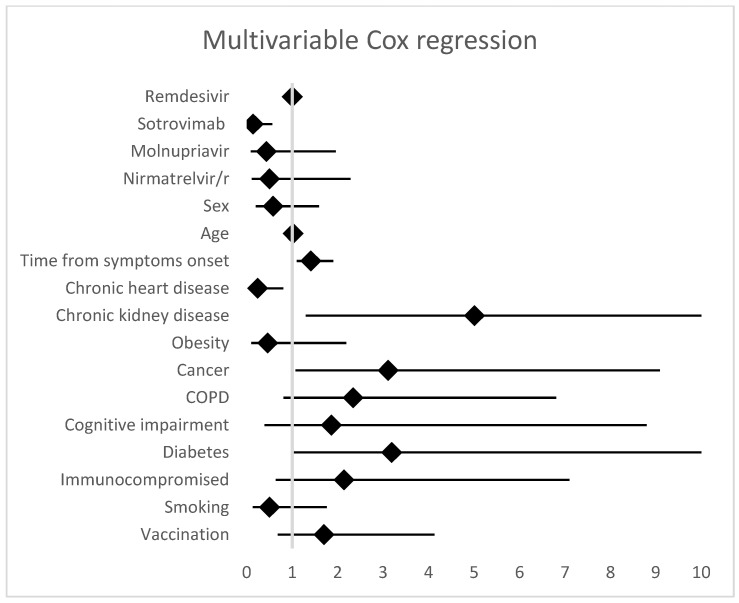
Multivariable Cox regression analysis. Predictors of composite outcome (28-day hospitalization and/or death related to COVD-19).

**Table 1 viruses-15-00438-t001:** Baseline characteristics of the study population, divided into the four treatment groups.

	TOTAL (*n* = 781)	RMD (*n* = 142)	SOT (*n* = 314)	MOL (*n* = 205)	NRM/r (*n* = 120)	*p*-Value
Sex (*n*, %)						
-- Male	394 (50.4)	59 (41.6)	166 (52.9)	118 (57.6)	51 (42.5)	
-- Female	387 (49.6)	83 (58.5)	148 (47.1)	87 (42.4)	69 (57.5)	0.005
Age (median, IQR)	66.9 (52.4–77.9)	67.4 (52–78.9)	64.7 (50.2–77.8)	68.9 (57.3–79.9)	66.9 (50.3–75.6)	0.014
Vaccination (*n*, %)						
-- None	108 (13.8)	17 (12)	64 (20.4)	24 (11.7)	3 (2.5)	
-- One dose	14 (1.8)	2 (1.4)	8 (2.6)	3 (1.5)	1 (0.8)	
-- Full schedule	144 (18.4)	24 (16.9)	67 (21.3)	46 (22.4)	7 (5.8)	
-- Booster	515 (65.9)	98 (79.7)	175 (55.7)	132 (64.4)	109 (90.8)	<0.001
Time from symptoms onset to treatment (median days, IQR)	3 (2–4)	4 (2–5)	4 (3–5)	3 (2–4)	3 (2–3)	<0.001
Obese (*n*, %)	178 (22.8)	26 (18.3)	62 (19.6)	61 (29.8)	29 (24.2)	0.027
Pregnant (*n*, %)	2 (0.3)	2 (1.4)	0	0	0	--
CKD (*n*, %)	75 (9.6)	8 (5.6)	47 (15)	16 (7.8)	4(3.3)	<0.001
CHD (*n*, %)	404 (51.7)	79 (55.6)	155 (49.4)	113 (55.1)	57 (47.5)	0.360
Cancer (*n*, %)	189 (24.2)	42 (29.6)	85 (27.1)	27 (13.2)	35 (29.2)	<0.001
COPD (*n*, %)	188 (24.1)	33 (23.2)	67 (21.3)	61 (29.8)	27 (22.5)	0.161
Cognitive impairment (*n*, %)	73 (9.4)	12 (8.5)	36 (11.5)	9 (4.4)	16 (13.3)	0.018
Stroke (*n*, %)	24 (3.1)	4 (2.8)	13 (4.1)	4 (2)	3 (2.5)	0.580
Diabetes (*n*, %)	141 (18.1)	29 (20.4)	53 (16.9)	33 (16.1)	26 (21.7)	0.501
Immunocompromised (*n*, %)	282 (36.1)	51 (35.9)	159 (50.6)	26 (12.7)	46 (38.3)	<0.001
Current or former smoker (*n*, %)	144 (18.4)	44 (31)	43 (13.7)	40 (19.5)	17 (14.2)	<0.001

Legend: RMD: remdesivir; SOT: sotrovimab; MOL: molnupiravir; NRM/r: nirmatrelvir/ritonavir; IQR: interquartile range; CKD: chronic kidney disease; CHD: chronic heart disease; COPD: chronic obstructive pulmonary disease.

## Data Availability

Data are available from the authors upon reasonable request.

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
