# Peer review of "Safety and Efficacy of Outpatient Treatments for COVID-19: Real-Life Data from a Regionwide Cohort of High-Risk Patients in Tuscany, Italy (the FEDERATE Cohort)"

_viruses, 2023, doi:10.3390/v15020438_

Round 1

Reviewer 1 Report

Authors presents real life data on safety and efficacy of COVID-19 outpatient early treatment from a wide cohort of 781 patients enrolled from January to March 2022 in 11 ID units all from Tuscany region in central Italy (the FEDERATE cohort). This in an interesting clinical research study using real life data from Italy that confirm that all used regimens share good safety and efficacy. Sotrovimab mnonoclonal antibody showed a reduced risk of progression versus remdesevir. No significant differences of outcome were observed in preventing 28-day hospitalization and death among patients treated with remdesivir, molnupiravir or nirmatrelvir/ritonavir 

Major concerns

Epidemiological data on the current VoCs circulating in Tuscany at the time of recruiting should be added in the summary and in the result section. 

The discussion on the VoC epidemiological data need to be more detailed. Now it appears to be too much minimalistic. 

The weight of the different criteria for early therapy eligibility on clinical outcome, should be added in the discussion. Particularly, the different trends of the two main determinants, chronic heart disease and chronic kidney disease, is noteworthy to be widely discussed. 

Finally, in the discussion section data on the maintaining efficacy of antiviral drugs on all known VoCs should be integrated 

Minor concerns

The following sentence in the abstract need to be reformulated: In the first Italian multicentric study presenting real-life data on COVID-19 47 early treatments.

Author Response

Authors presents real life data on safety and efficacy of COVID-19 outpatient
early treatment from a wide cohort of 781 patients enrolled from January to March 2022 in
11 ID units all from Tuscany region in central Italy (the FEDERATE cohort). This in an
interesting clinical research study using real life data from Italy that confirm that all used
regimens share good safety and efficacy. Sotrovimab monoclonal antibody showed a
reduced risk of progression versus remdesevir. No significant differences of outcome were
observed in preventing 28-day hospitalization and death among patients treated with
remdesivir, molnupiravir or nirmatrelvir/ritonavir 
We thank the Reviewer for taking the time to read and comment on our manuscript.
Point-by-point responses are provided below

Major concerns
Epidemiological data on the current VoCs circulating in Tuscany at the time of recruiting
should be added in the summary and in the result section. 
The discussion on the VoC epidemiological data need to be more detailed. Now it appears
to be too much minimalistic. 
We retrieved further epidemiological data about VoCs circulating in Tuscany at the
time of recruiting from the monthly updates by the National Institute of Health. In the
revised version we included more details in the summary and discussion
The weight of the different criteria for early therapy eligibility on clinical outcome, should be
added in the discussion. Particularly, the different trends of the two main determinants,
chronic heart disease and chronic kidney disease, is noteworthy to be widely discussed.
Thanks for pointing this out. We added some comments in the discussion about the
role of specific comorbidities on the outcome

Finally, in the discussion section data on the maintaining efficacy of antiviral drugs on all
known VoCs should be integrated 
A mention about maintained efficacy of antiviral drugs against all known VoCs was
already present in the discussion. However, we furtherly expanded this point
according to referee’s comment

Minor concerns
The following sentence in the abstract need to be reformulated: In the first Italian
multicentric study presenting real-life data on COVID-19 47 early treatments.
We corrected the sentence.

Reviewer 2 Report

The putative aim of this manuscript is to retrospectively evaluate different anti-COVID-19 treatments in order to evaluate their efficacy and tolerability in a large Italian population. There are already some works in the literature with similar objectives, however their small number in relation to the importance of the pathology means that this manuscript also deserves to be considered for publication by virus. The results obtained partly align with what we already know, reinforcing the data of the existing literature and this aspect is fundamental in relation to a pathology of which we still know very little and which is in continuous transformation.

However, I have some serious concerns that I would like to be dispelled.

Major revisions:

The text lacks a summary table of the symptoms that the patients displayed, as well as a thorough review of them in the discussion. In fact, the manifestations of COVID-19 can vary (DOI: 10.3390/ijms23169136) and be, for example, pulmonary, cardiological, gynecological (in the female population). Synthesizing them is not only important for descriptive purposes, but also for investigative purposes. In fact, a question I ask myself is: did the authors try to evaluate a correlation between specific clinical manifestations and disease progression? Or again between specific symptom and value/tolerability of the drug? In the multivariate analysis the possibility of evaluating this aspect should be considered, trying to investigate the possibility that a particular symptom or the set of two or more specific symptoms may have a significant value.

Another interesting aspect that the authors should explore is related to the vaccine variant. In fact, in Italy the vaccination campaign over time has made use of various vaccines (DOI: 10.3390/ijerph192013167). Did the authors investigate this aspect? If yes, what data has emerged? Even if negative, they must be reported in the text. Furthermore, also in this case, the type of vaccine and the number of doses administered should be included in the multivariate analysis.

It is also not clear to this reviewer why both vaccinated and unvaccinated patients are included in the results in the SOT, RMD, MOL and NRM/r groups. These two classes should be independent of each other.

I therefore propose to modify the discussion and the conclusions also in the light of the new results.

There are punctuation errors in the text that need to be corrected.

Author Response

REVIEWER 2
The putative aim of this manuscript is to retrospectively evaluate different anti-COVID-19
treatments in order to evaluate their efficacy and tolerability in a large Italian population.

There are already some works in the literature with similar objectives, however their small number in relation to the importance of the pathology means that this manuscript also deserves to be considered for publication by virus. The results obtained partly align with what we already know, reinforcing the data of the existing literature and this aspect is fundamental in relation to a pathology of which we still know very little and which is in continuous transformation.
However, I have some serious concerns that I would like to be dispelled.

We thank the Reviewer for the criticism and constructive comments. Point-by-point responses are provided below

Major revisions:
The text lacks a summary table of the symptoms that the patients displayed, as well as a
thorough review of them in the discussion. In fact, the manifestations of COVID-19 can vary (DOI: 10.3390/ijms23169136) and be, for example, pulmonary, cardiological, gynecological (in the female population). Synthesizing them is not only important for descriptive purposes, but also for investigative purposes. In fact, a question I ask myself is: did the authors try to evaluate a correlation between specific clinical manifestations and disease progression? Or again between specific symptom and value/tolerability of the drug? In the multivariate analysis the possibility of evaluating this aspect should be considered, trying to investigate the possibility that a particular symptom or the set of two or more specific symptoms may have a significant value.

This is a very interesting point. The suggestion that a particular symptom or the set of two or more specific symptoms may impact on the outcome is attractive but it is beyond the aim of our study. According to inclusion criteria, study population was made by subjects with mild to moderate COVID-19, thereby the pattern of symptoms might vary but the disease severity was homogeneous within the study population. However, we discuss this point among the limitations of the study Another interesting aspect that the authors should explore is related to the vaccine variant. In fact, in Italy the vaccination campaign over time has made use of various vaccines (DOI: 10.3390/ijerph192013167). Did the authors investigate this aspect? If yes, what data has emerged? Even if negative, they must be reported in the text. Furthermore, also in this case, the type of vaccine and the number of doses administered should be included in the
multivariate analysis.
We agree with the referee. Vaccination status was categorized as ‘no vaccinated’
(none or incomplete primary schedule) or ‘vaccinated’ (complete primary schedule +/- booster dose) for the multivariable analysis and the propensity score estimation. We included this information in the text. As for the type of vaccine we were not able to retrieve this information and we include this point among the limitations of the study
It is also not clear to this reviewer why both vaccinated and unvaccinated patients are included in the results in the SOT, RMD, MOL and NRM/r groups. These two classes should be independent of each other.

According to the aim of the study, the population was grouped by antiSARS-CoV-2 treatment received. Vaccination status, as well as other demographic and clinical features, was considered as an independent variable in both multivariable Cox regression model and propensity score analysis, leading to weight their potential impact on the outcome.

I therefore propose to modify the discussion and the conclusions also in the light of the new results.
Following all the comments received in the revision process, the discussion was
enriched with data and limitations suggested by the referees.

There are punctuation errors in the text that need to be corrected.
The text was carefully revised and the errors corrected. Thanks

Reviewer 3 Report

I was invited to revise the paper "entitled Safety and efficacy of outpatient treatments for COVID-19: real life data from a regionwide cohort of high risk patients in Tuscany, Italy (the FEDERATE cohort)". It was a cohort study performed in Tuscany Region, Italy, aimed to evaluate safety and efficacy of the four Covid treatment regimens (SOT, RMD, NRM/r and MOL) available in Italy from 11 Infectious Diseases Unit operating in the Tuscany Region, Italy.

The topic is very important and focus on a crucial point of Covid-19 patients care.

I have some observations:

- How did Authors handled eventual missing data? Did they performed multiple imputation?

- About Kaplan-Meier analysis, i suggest to report also median survival time with relative SE. In addition, I suggest to perform pairwise comparison between study groups;

- Authors should better define the "Obese" variable;

- About Ps matching, it is unclear if Authors performed the matching procedure before or after the survival analysis. In particular, in many variables the standardized mean diffewrence is higher than 10%, showing a bad balance among groups. In addition, afterPS score, Authors should match study group patients, removing patients that did not allow good balance and re-run analysis only in the matched population;

- As supplementary materials, Authors should analyze separately all study outcomes, reporting survival analysis of each ones.

Author Response

I was invited to revise the paper entitled "Safety and efficacy of outpatient treatments for COVID-19: real life data from a regionwide cohort of high risk patients in Tuscany, Italy (the FEDERATE cohort)". It was a cohort study performed in Tuscany Region, Italy, aimed to evaluate safety and efficacy of the four Covid treatment regimens (SOT, RMD, NRM/r and MOL) available in Italy from 11 Infectious Diseases Unit operating in the Tuscany Region, Italy.
The topic is very important and focus on a crucial point of Covid-19 patients care.
We thank the Reviewer for taking the time to read and comment on our manuscript.
Point-by-point responses are provided below
I have some observations:
- How did Authors handled eventual missing data? Did they performed multiple
imputation?
Thank you for pointing this out. Patients without outcome data were excluded from the analysis, as specified in the Methods. We added a table in the Supplementary
materials with the demographic and clinical features of the 140 subjects excluded from the analysis. As for the study population, we had no missing data, since they were retrieved from prescription CRF that not allowed empty items
- About Kaplan-Meier analysis, I suggest to report also median survival time with relative SE. In addition, I suggest to perform pairwise comparison between study groups;
We agree with the referee. In the revised version we reported average time at risk to event, computed as the time to the first event or day 28, whichever was earlier, and relative SE, as well as the results of pairwise comparison of survival curves of
treatment groups

- Authors should better define the "Obese" variable;
Obese was defined by a body-mass index [the weight in kilograms divided by the square of the height in meters] ≥30. We better specified this point in the Methods
- About Ps matching, it is unclear if Authors performed the matching procedure before or after the survival analysis. In particular, in many variables the standardized mean difference is higher than 10%, showing a bad balance among groups. In addition, after PS score, Authors should match study group patients, removing patients that did not allow good balance and re-run analysis only in the matched population; In fact we did not performed any matching procedure, as we prefer using inverse probability of treatment weighting (IPTW) to obtain unbiased estimates of average treatment effect by propensity score analysis.
Inverse probability of treatment weighting (IPTW) uses weights based on the
propensity score to create a synthetic sample in which the distribution of measured baseline covariates is independent of treatment assignment [Austin PC. An Introduction to Propensity Score Methods for Reducing the Effects of Confounding
In Observational Studies. Multivariate Behav Res. 2011 May;46(3):399-424].
We decided to use IPTW rather than matched propensity score analysis because
this method allows to maintain the entire study population, without necessity to
removing patients with poor balance
After weighing by the inverse probability of treatment, standardized differences of baseline covariates showed an acceptable balance (<0.2)
We better specified this point in the Methods
- As supplementary materials, Authors should analyze separately all study outcomes, reporting survival analysis of each ones.
Thank you for this suggestion. Study outcome include hospitalization and death. It should be considered that all 18 patients that met the composite endpoint were hospitalized (including 3 that subsequently died), while only 3 patients died. Separate analysis are likely to be poorly informative (“hospitalization” outcome overlap the composite outcome; the very low number of event prevents meaningful analysis on “death” outcome). However, we will be happy to include these analysis in the supplementary materials if referee and editor deem necessary

Round 2

Reviewer 2 Report

The modifications and explanations provided by the authors are considered exhaustive. Congratulations to the authors for the work done. If the authors deem it appropriate, I would ask to enrich the references with the previously suggested manuscripts.

Author Response

We thank the reviewer for his/her efforts and consideration. We added one of the suggested references. 

Reviewer 3 Report

Authors addressed all comments

Author Response

We thank the reviewer for his/her acknowledgment